# A Critical Issue in Lung Cancer Cytology and Small Biopsies: DNA and RNA Extraction from Archival Stained Slides for Biomarker Detection through Real Time PCR and NGS—The Experience in Pathological Anatomy Unit

**DOI:** 10.3390/diagnostics13091637

**Published:** 2023-05-05

**Authors:** Giuseppa Zannini, Ilaria Tedesco, Immacolata Cozzolino, Marco Montella, Eduardo Clery, Carminia Maria Della Corte, Floriana Morgillo, Marina Accardo, Renato Franco, Federica Zito Marino

**Affiliations:** 1Pathology Unit, Department of Mental and Physical Health and Preventive Medicine, University of Campania “Luigi Vanvitelli”, Via L. Armanni 5, 80138 Naples, Italy; giuseppa.zannini@unicampania.it (G.Z.); ilaryted@gmail.com (I.T.); coimma73@gmail.com (I.C.); marco.montella@unicampania.it (M.M.); eduardoclery87@gmail.com (E.C.); marina.accardo@unicampania.it (M.A.); federica.zitomarino@unicampania.it (F.Z.M.); 2Medical Oncology, Department of Precision Medicine, University of Campania “Luigi Vanvitelli”, Via S. Pansini 5, 80131 Naples, Italy; carminiamaria.dellacorte@unicampania.it (C.M.D.C.); floriana.morgillo@unicampania.it (F.M.)

**Keywords:** lung cancer, cytological lung samples, biomaterials, NGS, Real Time PCR, predictive biomarkers, precision medicine, molecular biology

## Abstract

The handling of biomaterials is crucial for precision medicine in advanced-stage lung patients with only cytology or small biopsies available. The main purpose of the study was to evaluate the quantity and quality of nucleic acids extracted from mixed stained slides (MSSs), including H&E, IHC and FISH, compared to the extraction from unstained slides (USs). A series of 35 lung adenocarcinoma surgical samples was selected to set up the method and the technical approach was validated in a series of 15 small biopsies and 38 cytological samples. DNA extracted from MSSs was adequate in all samples and the Real Time PCR was successful in 30/35 surgical samples (86%), 14/15 small biopsies (93%), and 33/38 cytological samples (87%). NGS using DNA extracted from MSSs was successful in 18/35 surgical samples (51%), 11/15 small biopsies (73%), and 26/38 cytological samples (68%). RNA extracted from MSSs was unsatisfactory in all cases showing an inadequate degree of fragmentation. Our technical approach based on the recovery of stained slides could represent a strategic way forward for DNA-based biomarker testing in lung cancer cases without biomaterials. The RNA extracted from MSSs did not represent a successful approach.

## 1. Introduction

Over the last decade, the increasing knowledge of cancer molecular biology has led to the development of personalized therapies for the treatment of oncologic patients. In particular, the molecular landscape of non-small cell lung carcinoma (NSCLC) is rapidly evolving, and the treatment of this cancer has progressively become more and more biomarker-driven [1]. Among the identified oncogene driver mutations, some must necessarily be tested in clinical practice, being recommended for specifically approved target therapy, including EGFR mutations, ALK rearrangement, ROS1 rearrangement, BRAF mutations, MET exon 14 skipping mutations [2].

In this broad molecular framework, the amount of biomaterial for routine testing is a priority to be managed more carefully, since up to 80% of NSCLC patients are in the advanced, not operable stage and have cytology samples or small biopsies as the only available biomaterial [3]. The numerous changes in the field of precision oncology require optimizing the biomaterial to perform biomarker testing for treatment decisions. In this study, we propose a technical approach to re-use the archival biomaterial to maximize available material for molecular analysis. The main aim of the study was to evaluate the possible use of archival material, especially hematoxylin and eosin (H&E), immunohistochemistry (IHC), and fluorescence in situ hybridization (FISH) slides, to extract DNA and RNA. At first, we set up the technique using sections obtained from a series of lung adenocarcinoma surgical samples, containing a greater amount of available biomaterial. Moreover, we validated the technical approach in sections from a series of small biopsies and cytological lung adenocarcinomas. Thus, the quality and quantity of the extracted nucleic acids were evaluated and compared with those extracted from unstained sections using a standard protocol. In addition, the DNA and RNA extracted from stained slides were used to perform real-time polymerase chain reaction (PCR) and next-generation sequencing (NGS) to evaluate the yield.

## 2. Materials and Methods

### 2.1. Design of the Study

We retrospectively reviewed archival surgical samples, small biopsies, and cytological samples of lung adenocarcinoma with available material diagnosed at the Pathology Unit of the University of Campania Luigi Vanvitelli from 2019 to 2023 (Appendix A). All samples were stored in the same condition in our archive. The study design included the following:(i)Setup of the technique in a series of 35 surgical samples:

For each sample, five different DNA and RNA extraction were performed starting from different types of stained slides 4–5 µm to evaluate the yield of the extraction, particularly unstained slides (USs) used as a gold standard, H&E-stained slides, IHC slides, FISH slides and the last setting including all different types of stained slides (Figure 1). The H&E-stained slides, IHC slides, and FISH slides were prepared from FFPE tissue blocks. The DNA and RNA extraction from histologic samples was performed both through an automatic extraction and a column-based approach. The workflow of the technical setup is summarized in Figure 1.

(ii)Validation of the technique in a series of 15 small biopsies and 38 cytological samples:

For each sample, two different DNA and RNA extractions were performed, one starting from USs used as a gold standard and the other from mixed stained slides setting (MSSs), including H&E, IHC, and FISH (Figure 1). All archival slides were digitalized before the DNA and RNA extraction. The cytological samples selected were characterized by moderate (between 100 and 500 representative cells in the sample) or high cellularity (>500 representative cells in the sample). The DNA and RNA extraction from cytologic samples and small biopsies were performed through a column-based approach.

### 2.2. Immunohistochemistry and Immunocytochemistry

All immunohistochemistry and immunocytochemistry were performed on the BenchMark Ultra automated staining platform from Ventana Medical Systems. Both the OptiView Universal DAB Detection Kit and UltraView Universal DAB Detection Kit were used depending on the antibody and its protocol. The following antibodies from Ventana Medical Systems were used according to validated protocols: anti-Cytokeratin 7 (SP52) Rabbit Monoclonal Primary Antibody; anti-Thyroid Transcription Factor-1 (8G7G3/1) Mouse Monoclonal Primary Antibody; anti-ALK (D5F3) Rabbit Monoclonal Primary Antibody; anti-ROS1 (SP384) Rabbit Monoclonal Primary Antibody; PD-L1 (SP263) Assay.

### 2.3. Fluorescence in Situ Hybridization

The FISH assay was performed using the Bond FISH kit on the automated Bond system (Leica Biosystems, Vista, CA, USA). FISH analysis was performed according to our protocol previously reported [4]. We have used the follow commercially available probes: ZytoLight^®^ SPEC ROS1 Dual Color Break Apart Probe; ZytoLight^®^ SPEC RET Dual Color Break Apart Probe; ZytoLight^®^ SPEC ALK Dual Color Break-Apart Probe; ZytoLight^®^ SPEC NTRK1 Dual Color Break Apart Probe; ZytoLight^®^ SPEC NTRK2 Dual Color Break Apart Probe; ZytoLight^®^ SPEC NTRK3 Dual Color Break Apart Probe (ZytoVision, GmbH, Bremerhaven, Germany).

### 2.4. Digital Scan

All archival slides were digitally scanned by APERIO LV1 (Leica Biosystems, Vista, CA, USA) and saved to a digital database. Aperio ImageScope Software (Leica Biosystems, Vista, CA, USA) was used for remote access and viewing of scanned cases.

### 2.5. Tissue Dissection for DNA and RNA Extraction

For each sample, an enrichment of neoplastic cells was performed through automatic dissection using AVENIO MilliSect System (Roche, Pleasanton, CA, USA) according to the protocol previously published [5].

### 2.6. DNA and RNA Extraction

#### 2.6.1. Automatic Extraction for Histological Samples

Genomic DNA and total RNA were isolated starting from both USs and MSSs of histological samples using the MagCore^®^ Automated Extraction Instruments following the manufacturer’s instructions. DNA and RNA were eluted in 60 µL of DNAse and RNAse-free water supplied by the kit.

#### 2.6.2. Column-Based Extraction for Histological and Cytological Samples

Column-based extraction was used for the isolation of genomic DNA and total RNA from MSSs of histological, small biopsies, and cytological samples using the AllPrep DNA/RNA Mini Kit (Qiagen, Valencia, CA, USA) according to the manufacturer’s instructions. DNA and RNA were eluted in 30 µL of DNAse and RNAse-free water (Thermo Fisher Scientifics, Waltham, MA, USA)

### 2.7. Assessment of DNA and RNA Quality

The DNA and RNA were quantified with NanoDrop 2000c (Thermoscientific). The fragmentation was evaluated through the amplification of the long and the short fragment by Real Time PCR using Myriapod^®^ NGS Cancer panel DNA kit and Myriapod^®^ NGS Cancer panel RNA kit. The fragmentation was calculated as the ratio between the concentration (ng/µL) of the longer length and that of the shorter length amplicon of the nucleic acids. The degree of DNA fragmentation was calculated by Real Time PCR as the ratio between the amplification of longer fragments evaluated in the HEX channel and shorter fragments in the FAM channel. The degree of RNA fragmentation was calculated by Real Time PCR as the ratio between the amplification of longer fragments evaluated in the FAM channel and shorter fragments in the HEX channel. The fragmentation index was evaluated as follows: >0.3 corresponded to a low degree of fragmentation, ranging between 0.1–0.3 to medium and <0.1 to high.

### 2.8. Real Time Polymerase Chain Reaction

The DNA extracted from both USs and MSSs of all cases was used for EGFR and BRAF mutations detection by Real Time PCR using the kits EasyPGX^®^ ready EGFR and EasyPGX^®^ ready BRAF following the manufacturer’s instructions. The RNA extracted from both USs and MSSs of all cases was used for the detection of lung fusions by Real Time PCR using the kit EasyPGX^®^ ready ALK, ROS1, RET, MET following the manufacturer’s instructions. ΔCq is calculated as the difference between the Cq value of the gene mutation and the Cq value of the gene control. The sample is classified as mutated if the ΔCq value is equal to or lower than the cut-off of the specific target, as reported in the manufacturer’s instructions.

### 2.9. Next-Generation Sequencing

The DNA and RNA extracted from both USs and MSSs were used for NGS analyses. Only the nucleic acids with a fragmentation index > 0.1 were tested by NGS. The libraries were prepared using the Myriapod^®^ NGS Cancer panel DNA and the Myriapod^®^ NGS Cancer panel RNA for Illumina iSeq™100 platform (Illumina, San Diego, CA, USA) according to the manufacturer’s protocol. The normalized library pool was sequenced on the Illumina Iseq100. Sequencing data analysis was analyzed locally by the dedicated Myriapod NGS Data Analysis Software.

## 3. Results

### 3.1. Evaluation of DNA and RNA Extracted from Archival Material

#### 3.1.1. Histological Control Samples

The DNA extracted from MSSs was successful, suggesting that no reagent used in the previous stains can affect the DNA extraction. In all 35 cases, the quantity of DNA extracted from -showed a decrease compared to that extracted from USs probably due to the loss of starting tumor cells because of the technical phases of removing the coverslips. The mean concentration of DNA extracted from MSSs was 122.8 ng/µL compared to 276.9 ng/µL from USs. In detail, the decrease in DNA quantity was variable, particularly 14 out of 35 cases (40%) showed a decrease < 100 ng/µL, 18 out of 35 cases (51%) by 100–300 ng/µL, three out of 35 cases (9%) a decrease > 300 ng/µL. In all 35 cases, the quality of DNA extracted from MSSs showed a decrease compared to that from USs. The mean of the fragmentation index of DNA extracted from MSSs was 0.11 compared to 0.22 from USs. In detail, the increase of DNA fragmentation was variable, particularly 24 out of 35 cases (68%) showed a low increase, nine out of 35 cases (26%) moderate increase, two out of 35 cases (6%) high increase of fragmentation. All data are summarized in Table 1.

The RNA extracted from MSSs was unsatisfactory, suggesting that some staining process reagents can affect RNA extraction. In all 35 cases, the quantity of RNA extracted from MSSs showed a decrease compared to that extracted from USs, probably due to the loss of starting tumor cells because of the technical phases of removing the coverslips. The mean concentration of RNA extracted from MSSs was 61.1 ng/µL compared to 146.9 ng/µL from USs. In detail, the decrease in RNA quantity was variable, particularly 25 out of 35 cases (71%) showed a decrease by <100 ng/µL, 10 out of 35 cases (29%) by 100–300 ng/µL, and no cases showed a decrease >300 ng/µL. In all 35 cases, the quality of RNA extracted from MSSs was unsatisfactory, and all cases have an inadequate degree of fragmentation. The mean of the fragmentation index of RNA extracted from MSSs was 0 compared to 0.16 from USs. All data are summarized in Table 2.

As expected, in the cohort of histological samples, DNA and RNA extraction yield was higher with the column-based than with the automated method.

#### 3.1.2. Small Biopsies and Cytological Samples

The mean concentration of DNA extracted from MSSs of small biopsies cases was 50.0 ng/µL. In detail, the quantity of DNA extracted was <30 ng/µL in three out of 15 small biopsies (20%), between 30–50 ng/µL in five cases (34%) and e > 50 ng/µL in seven cases (46%). The mean of the fragmentation index of DNA extracted from MSSs was 0.22. In detail, the DNA degree of fragmentation was high in four out of 15 cases (27%), moderate in four cases (27%), and low in seven cases (46%). All data are summarized in Table 3.

The mean concentration of RNA extracted from MSSs of small biopsies cases was 32.4 ng/µL. In detail, the quantity of RNA extracted was <30 ng/µL in six out of 15 (40%), between 30–50 ng/µL in eight cases (53%) and e > 50 ng/µL in only one case (7%). The quality of RNA extracted from mixed settings was unsatisfactory, and all small biopsies cases have an inadequate degree of fragmentation (Figure 2). All data are summarized in Table 4.

The mean concentration of DNA extracted from MSSs of cytological cases was 19.8 ng/µL. In detail, the quantity of DNA extracted from MSSs setting of cytological samples was <10 ng/µL in nine out of 38 (24%), between 10–30 ng/µL in 25 cases (65%) e > 30 ng/µL in four cases (11%). The mean of the fragmentation index of DNA extracted from MSSs was 0.14. In detail, the DNA degree of fragmentation from MSSs of cytological samples was high in 12 out of 38 cases (32%), moderate in 24 cases (63%), and low in two cases (5%). All data are summarized in Table 5.

The mean concentration of RNA extracted from MSSs of cytological cases was 24.0 ng/µL. In detail, the quantity of RNA extracted from the mixed setting of cytological samples was <10 ng/µL in six out of 38 (16%), between 10–30 ng/µL in 22 cases (58%) e > 30 ng/µL in 10 cases (26%). The quality of RNA extracted from the mixed setting was unsatisfactory; all cytological cases have an inadequate degree of fragmentation. All data are summarized in Table 6.

The extraction of cytologic samples and small biopsies was performed exclusively through the column-based method, since the amount of the biomaterial was limited, and the automatic extraction was not recommendable.

### 3.2. Real Time PCR and NGS Using Nucleic Acids Extracted from Archival Material

#### 3.2.1. Histological Control Samples

In all 35 cases, the DNA extracted from the USs was used for Real Time PCR analysis showing successful results. The Real Time PCR performed using the DNA extracted from MSSs was successful in 30 out of 35 cases (Table 1). According to the degree of fragmentation, DNA extracted from MSSs was eligible to perform an NGS assay in 18 out of 35 cases (51%) compared to that extracted from the USs adequate in 26 out of 35 (74%) (Table 1 and Appendix A).

The RNA extracted from the USs was used for Real Time PCR analysis showing successful results in 32 out of 35 cases. The Real Time PCR performed using the RNA extracted from MSSs was unsatisfactory in all cases (Table 2). According to the degree of fragmentation, RNA extracted from the MSSs setting was not eligible to perform NGS assay compared to that extracted from the USs adequate in 23 out of 35 (66%) (Table 2).

#### 3.2.2. Small Biopsies and Cytological Samples

The Real Time PCR performed using DNA extracted from MSSs of small biopsies cases was successful in 14 out of 15 cases (93%) (Table 3). According to the degree of fragmentation, DNA extracted from MSSs of small biopsies was eligible to perform NGS assay in 11 out of 15 cases (73%) (Table 3 and Appendix A).

The Real Time PCR performed using DNA extracted from MSSs of cytological samples was successful in 33 out of 38 cases (87%) (Figure 3). According to the degree of fragmentation, DNA extracted from MSSs of cytological samples was eligible to perform NGS assay in 26 out of 38 cases (68%) (Table 5 and Appendix A).

The Real Time PCR performed using the RNA extracted from MSSs both of small biopsies and cytological samples was unsatisfactory in all cases (Table 4 and Table 6). According to the degree of fragmentation, the RNA extracted from MSSs both of small biopsies and cytological samples was not eligible to perform NGS assay.

## 4. Discussion

The handling of small biopsies and cytological samples is one of the main problems of the Pathological Anatomy Laboratory in the daily clinical practice of lung cancer patients. More and more molecular responses are required on small specimens; thus, sample optimization is mandatory for defining the appropriate treatment of NSCLCs.

Previous data showed that the simultaneous detection of multiple genetic aberrations is needed to make the best use of the available biomaterial [4,6]. Our previous study demonstrated the use of multiplex FISH to detect simultaneously ALK-R and ROS1-R, ensuring both the detection of the rearrangement with FISH assay considered the gold standard and the savings of small samples [4]. In the last time, other gene fusions have been identified as actionable genetic alterations in NSCLC, i.e., NTRK fusions, and the guidelines recommend their detection through IHC and confirmed by NGS, overcoming the FISH approach historically used for the rearrangement identification. Recently, also the detection of MET ex14 skipping has become a routine test in advanced NSLCs, and the RNA-based assays have a higher accuracy compared to DNA-based tests to detect this aberration [7,8,9]. In the NSCLC clinical context, DNA extraction is no longer enough, but RNA extraction has also become mandatory.

In this view, the recovery of archival material to extract the nucleic acids could be a solution to have available biomaterial for molecular profiling of cases with small amounts of sample.

Previous studies have investigated the yield of DNA extracted from H&E, using different protocols of extraction. Although DNA extracted from H&E showed lower recovery and some degree of fragmentation compared to unstained starting samples, however, it has proven to be adequate to perform PCR with high efficiency [10,11]. Several studies evaluated DNA extracted also from archived Papanicolaou-stained smears of lung adenocarcinoma demonstrating that this stain does not influence DNA extraction, allowing for a successful mutational analysis by PCR-based and NGS [12,13,14].

To our knowledge, no previous study has systematically evaluated the yield of the DNA and RNA extraction from archival material, including IHC and FISH slides.

The main objective of our study was to analyze the recovery of the archival slides for the extraction of DNA and RNA, evaluating the quality and quantity of the extracted nucleic acids compared to the USs as starting biomaterial.

Our data demonstrated that DNA extracted from H&E, IHC, and FISH slides showed a decrease in the quantity probably due to the various technical steps that require the removal of the coverslip associated with a possible loss of biomaterial.

Although DNA extracted from MSSs showed a decrease in quality, about 73% of small biopsies and 83% of cytological samples were eligible for mutational analysis that reported successful results. The results of our study suggest that the performance of Real Time PCR and NGS using DNA extracted from archival slides are comparable to those performed with nucleic acids extracted from USs according to standard protocol.

To date, few data are reported regarding RNA extraction from H&E slides. Previous results demonstrated that the RNA extracted from H&E frozen sections, using RNase-free conditions, could ensure good integrity of the nucleic acid [10,15]. However, the RNA extracted from frozen sections is generally not part of the clinical practice workflow of NSCLC patients for molecular analysis. Trejo and colleagues described the use of a ligation-based targeted whole transcriptome expression profiling assay, TempO-Seq, using as starting biomaterial the H&E slides obtained from FFPE. They showed that this technique was able to detect highly precise and reproducible gene expression information also from H&E-stained slides, as long as the staining is performed using RNase-free reagents. However, the TempO-Seq is not sensitive to fragmentation since it is not based on RNA extraction and reverse transcription, since it uses directly lysing tissue scraped from slides as input for the annealing step [16].

In our series, the quantity of RNA extracted from mixed archival slides was reduced by 58.4% compared to that extracted from USs, probably due to the technical limits above described for DNA.

Regarding the RNA degree of fragmentation, our results show that the RNA extracted from archival slides does not have good integrity, suggesting that the reagents used in previous stainings have an influence on RNA degradation.

The quality of RNA extracted is a critical point for archival FFPE samples since the RNA degradation is more frequent than the samples compared to fresh or frozen tissue.

Several factors could affect the degradation of RNA, particularly the pre-analytical variables, the formalin fixation time, the tissue storage, and the cold or warm ischemia time could affect nucleic acids of FPPE specimens [17,18].

The assumptions of our study were based on the recovery of archival material both to optimize the NSCLC small samples and to perform orthogonal methods in cases with doubtful results with a small amount of material. Unfortunately, the quality of RNA extracted from stained slides has disappointed our expectations; however, our technical approach could ensure a significant advantage for DNA-based molecular analysis. In conclusion, our workflow proposing the reuse of archive material for molecular analysis could represent a valid approach in the clinical practice of NSCLCs with a limited biomaterial that otherwise could not be tested.

## Figures and Tables

**Figure 1 diagnostics-13-01637-f001:**
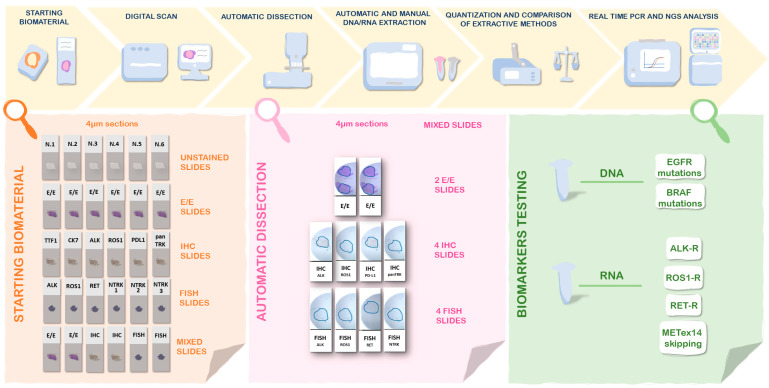
Workflow of the study: setup of the technical approach in lung cancer surgical samples.

**Figure 2 diagnostics-13-01637-f002:**
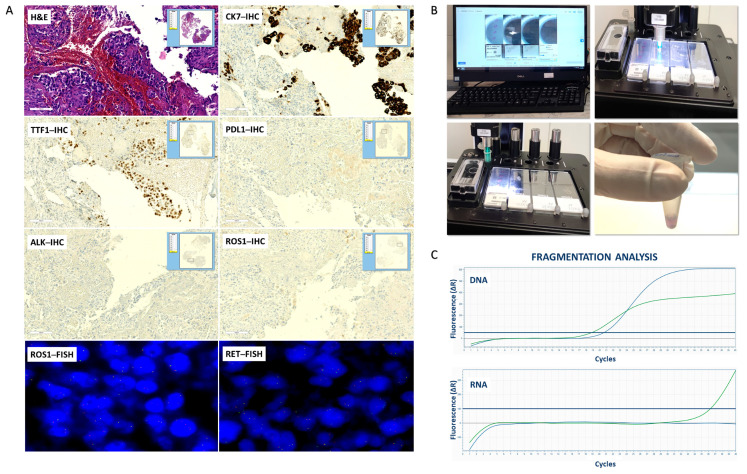
Workflow of the handling of small biopsies: digital scanning, automatic dissection, and fragmentation analysis. (**A**) Digital imaging of MSSs. Hematoxylin-eosin (H&E) stained slide (original magnification 20×); CK7, TTF1, PD–L1, ALK and ROS1 immunohistochemical (IHC) stained slides (original magnification 20×); ROS1 and RET fluorescence in situ hybridization (FISH) slides (original magnification 100×); (**B**) Automatic dissection; (**C**) Fragmentation analysis of DNA and RNA extracted from MSSs. DNA amplification of longer fragments (green line, HEX) and shorter fragments (blue line, FAM) by Real Time PCR. RNA amplification of longer fragments (blue line, FAM) and shorter fragments (green line, HEX) by Real Time PCR.

**Figure 3 diagnostics-13-01637-f003:**
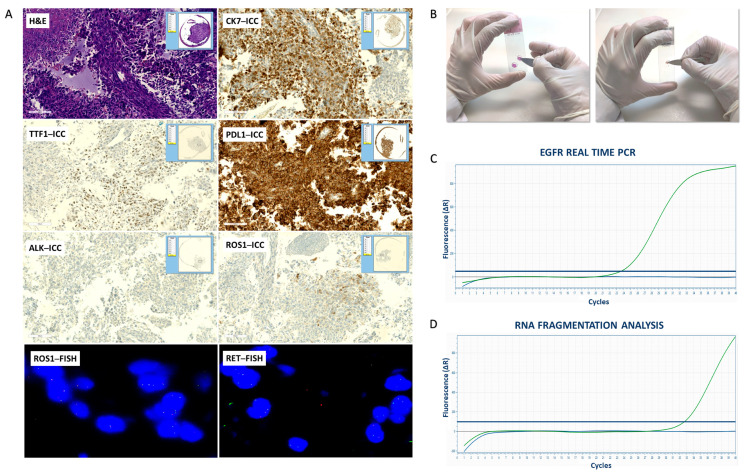
Workflow of the handling of cytological samples: digital scanning, manual dissection, and fragmentation analysis. (**A**) Digital imaging of MSSs. Hematoxylin-eosin (H&E) stained slide (original magnification 20×); CK7, TTF1, PD–L1, ALK and ROS1 immunocytochemical (ICC) stained slides (original magnification 20×); ROS1 and RET fluorescence in situ hybridization (FISH) slides (original magnification 100×); (**B**) Manual dissection; (**C**) EGFR Real Time PCR analyses of DNA extracted from MSSs. Control (green line, HEX) and EGFR L858R exon 21 mutation (blue line, FAM) by Real Time PCR. (**D**) Fragmentation analysis of RNA extracted from MSSs. RNA amplification of longer fragments (blue line, FAM) and shorter fragments (green line, HEX) by Real Time PCR.

**Table 1 diagnostics-13-01637-t001:** DNA extracted from USs and MSSs of histological samples.

DNA Histological Samples										
Case	USs (n. 10 Slides)					MSSs (*n*. 10 Slides)				
	[] (ng/µL)	A_260/280_	Degree of Fragmentation	Molecular Test Success		[] (ng/µL)	A_260/280_	Degree of Fragmentation	Molecular Test Success	
				Real Time PCR	NGS				Real Time PCR	NGS
1	217.3	1.9	0.38	+	+	74.3	1.7	0.21	+	+
2	344.5	1.8	0.26	+	+	120.6	1.7	0.14	+	+
3	189.5	1.8	0.44	+	+	56.2	1.6	0.35	+	+
4	215.2	1.8	0.33	+	+	59.8	1.7	0.23	+	+
5	138.4	1.9	0.27	+	+	54.5	1.8	0.11	+	+
6	220.3	1.7	0.12	+	+	137.7	1.6	0	-	NP
7	160.3	1.8	0.33	+	+	35.0	1.6	0.14	+	+
8	350.8	1.8	0.02	+	NP	283.8	1.6	0	-	NP
9	129.5	1.8	0.41	+	+	69.3	1.7	0.24	+	+
10	138.8	1.8	0.42	+	+	50.5	1.7	0.31	+	+
11	68.0	1.8	0.44	+	+	20.6	1.6	0.24	+	+
12	230.1	1.8	0.51	+	+	48.6	1.7	0.32	+	+
13	543.1	1.9	0.01	+	NP	266.5	1.6	0	+	NP
14	107.1	1.8	0.41	+	+	85.1	1.6	0.32	+	+
15	354.2	1.9	0.14	+	+	230.4	1.8	0	+	NP
16	360.5	1.9	0.14	+	+	132.9	1.8	0	+	NP
17	265.7	1.8	0.04	+	NP	112.1	1.7	0	+	NP
18	335.6	1.8	0.43	+	+	164.3	1.7	0.24	+	+
19	189.5	1.9	0.24	+	+	91.5	1.8	0.12	+	+
20	285.5	1.9	0.04	+	NP	124.0	1.8	0	+	NP
21	125.3	1.8	0.12	+	+	70.1	1.7	0	+	NP
22	301.4	1.8	0.04	+	NP	129.5	1.6	0	-	NP
23	264.5	1.9	0.11	+	+	229.4	1.8	0	+	NP
24	203.6	1.8	0.16	+	+	141.2	1.7	0	+	NP
25	833.9	1.8	0.13	+	+	153.4	1.6	0	+	NP
26	172.7	1.7	0.01	+	NP	121.5	1.6	0	-	NP
27	433.6	1.8	0.03	+	NP	179.0	1.7	0	+	NP
28	534.7	1.9	0.32	+	+	197.0	1.7	0.14	+	+
29	233.4	1.7	0.01	+	NP	113.6	1.6	0	+	NP
30	476.1	1.8	0.52	+	+	230.5	1.7	0.32	+	+
31	172.7	1.9	0.03	+	NP	151.9	1.7	0	+	NP
32	173.6	1.9	0.23	+	+	28.3	1.7	0.11	+	+
33	369.4	1.7	0.32	+	+	67.8	1.6	0.14	+	+
34	287.2	1.8	0.12	+	+	207.5	1.7	0	-	NP
35	264.7	1.9	0.33	+	+	59.3	1.8	0.21	+	+

[]: concentration; A: absorbance; USs: unstained slides; MSSs: mixed stained slides; +: performed with an adequate result; -: performed without an adequate result; NP: not performed.

**Table 2 diagnostics-13-01637-t002:** RNA extracted from USs and MSSs of histological samples.

RNA Histological Samples										
Case	USs (*n*. 10 Slides)					MSSs (*n*. 10 Slides)				
	[] (ng/µL)	A_260/280_	Degree of Fragmentation	Molecular Test Success		[] (ng/µL)	A_260/280_	Degree of Fragmentation	Molecular Test Success	
				Real Time PCR	NGS				Real Time PCR	NGS
1	35.4	1.8	0.32	+	+	16.9	1.4	0	-	NP
2	168.3	1.7	0.22	+	+	93.9	1.3	0	-	NP
3	132.3	1.7	0.21	+	+	61.3	1.2	0	-	NP
4	62.3	1.7	0.17	+	+	27.0	1.1	0	-	NP
5	50.1	1.8	0.04	+	NP	25.4	1.2	0	-	NP
6	181.6	1.7	0.03	+	NP	78.9	1.6	0	-	NP
7	56.4	1.6	0.21	+	+	22.3	1.3	0	-	NP
8	192.2	1.7	0.05	+	NP	93.6	1.6	0	-	NP
9	200.3	1.8	0.07	+	NP	80.9	1.6	0	-	NP
10	321.2	1.8	0.21	+	+	72.5	1.5	0	-	NP
11	43.8	1.7	0.03	+	NP	25.5	1.4	0	-	NP
12	129.8	1.7	0.21	+	+	36.8	1.3	0	-	NP
13	352.1	1.7	0.08	+	NP	162.1	1.6	0	-	NP
14	73.6	1.7	0.02	+	NP	35.7	1.3	0	-	NP
15	129.3	1.7	0.01	-	NP	63.9	1.5	0	-	NP
16	218.4	1.8	0	-	NP	67.8	1.5	0	-	NP
17	88.7	1.8	0.21	+	+	55.7	1.5	0	-	NP
18	91.2	1.7	0.23	+	+	79.0	1.6	0	-	NP
19	211.5	1.6	0.12	+	+	65.2	1.4	0	-	NP
20	140.5	1.6	0.34	+	+	63.3	1.4	0	-	NP
21	61.9	1.7	0.11	+	+	44.2	1.4	0	-	NP
22	100.3	1.6	0	+	NP	67.3	1.5	0	-	NP
23	268.3	1.6	0.24	+	+	58.6	1.5	0	-	NP
24	128.3	1.7	0	-	NP	51.6	1.5	0	-	NP
25	142.5	1.7	0.33	+	+	82.1	1.6	0	-	NP
26	132.2	1.7	0.17	+	+	46.6	1.5	0	-	NP
27	88.4	1.8	0.36	+	+	60.6	1.5	0	-	NP
28	179.3	1.7	0.22	+	+	77.6	1.5	0	-	NP
29	93.5	1.7	0.17	+	+	59.3	1.6	0	-	NP
30	342.7	1.6	0.24	+	+	115.4	1.5	0	-	NP
31	85.6	1.7	0.12	+	+	32.2	1.6	0	-	NP
32	44.2	1.8	0.15	+	+	15.6	1.4	0	-	NP
33	47.0	1.7	0.31	+	+	31.3	1.5	0	-	NP
34	428.2	1.7	0.33	+	+	129.8	1.4	0	-	NP
35	120.3	1.7	0.02	+	NP	38.9	1.5	0	-	NP

[]: concentration; A: absorbance; USs: unstained slides; MSSs: mixed stained slides; +: performed with an adequate result; -: performed without an adequate result; NP: not performed.

**Table 3 diagnostics-13-01637-t003:** DNA extracted from MSSs of small biopsies samples.

DNA Small Biobsies MSSs (*n*. 10 Slides)						
Case	[] (ng/µL)	A_260/280_	A_260/230_	Degree of Fragmentation	Molecular Test Success	
					Real Time PCR	NGS
1	40.7	1.6	0.4	0.21	+	+
2	72.4	1.6	0.4	0.14	+	+
3	65.6	1.5	0.4	0.02	+	NP
4	49.9	1.5	0.3	0.31	+	+
5	27.8	1.4	0.3	0	-	NP
6	70.3	1.7	0.7	0.04	+	NP
7	71.1	1.6	0.4	0.23	+	+
8	24.6	1.6	0.4	0.31	+	+
9	32.4	1.5	0.3	0.33	+	+
10	56.7	1.5	0.4	0.41	+	+
11	46.1	1.7	0.4	0.22	+	+
12	22.5	1.6	0.4	0.04	+	NP
13	43.2	1.7	0.5	0.33	+	+
14	67.8	1.5	0.7	0.41	+	+
15	58.3	1.6	0.3	0.32	+	+

[]: concentration; A: absorbance; MSSs: mixed stained slides; +: performed with an adequate result; -: performed without an adequate result; NP: not performed.

**Table 4 diagnostics-13-01637-t004:** RNA extracted from MSSs of small biopsies samples.

RNA Small Biobsies MSSs (*n*. 10 Slides)						
Case	[] (ng/µL)	A_260/280_	A_260/230_	Degree of Fragmentation	Molecular Test Success	
					Real Time PCR	NGS
1	22.8	1.7	0.4	0	-	NP
2	34.6	1.9	0.2	0	-	NP
3	32.3	1.5	0.5	0	-	NP
4	21.3	1.3	0.8	0	-	NP
5	60.2	1.6	0.6	0	-	NP
6	32.3	1.5	0.1	0	-	NP
7	25.8	1.2	0.6	0	-	NP
8	14.2	1.4	0.4	0	-	NP
9	21.0	1.4	0.3	0	-	NP
10	43.6	1.4	0.5	0	-	NP
11	35.6	1.4	0.3	0	-	NP
12	10.3	1.6	0.5	0	-	NP
13	34.8	1.5	0.5	0	-	NP
14	48.2	1.3	0.8	0	-	NP
15	48.5	1.4	0.7	0	-	NP

[]: concentration; A: absorbance; MSSs: mixed stained slides; -: performed without an adequate result; NP: not performed.

**Table 5 diagnostics-13-01637-t005:** DNA extracted from MSSs of cytological samples.

DNA Cytological MSSs (*n*. 10 Slides)						
Case	[] (ng/µL)	A_260/280_	A_260/230_	Degree of Fragmentation	Molecular Test Success	
					Real Time PCR	NGS
1	24.9	1.6	0.4	0.13	+	+
2	23.4	1.6	0.4	0.22	+	+
3	18.3	1.6	0.4	0.18	+	+
4	25.7	1.4	0.4	0.25	+	+
5	24.7	1.5	0.4	0.09	+	NP
6	21.8	1.5	0.3	0.11	+	+
7	17.5	1.7	0.3	0.04	+	NP
8	9.0	1.4	0.3	0.09	+	NP
9	8.2	1.5	0.3	0.07	+	NP
10	9.5	1.7	0.3	0.02	-	NP
11	9.7	1.4	0.2	0.14	+	+
12	56.5	1.4	0.3	0.26	+	+
13	6.7	1.6	0.3	0.21	+	+
14	7.8	1.4	0.3	0.12	+	+
15	11.2	1.6	0.3	0.27	+	+
16	16.3	1.1	0.4	0.41	+	+
17	6.9	1.5	0.5	0.31	+	+
18	16.4	1.4	0.2	0.21	+	+
19	9.4	1.3	0.3	0.24	+	+
20	11.2	1.6	0.4	0.16	+	+
21	10.0	1.5	0.4	0.04	+	NP
22	14.9	1.5	0.3	0.14	+	+
23	12.0	1.3	0.3	0.18	+	+
24	16.8	1.7	0.4	0.28	+	+
25	5.6	1.6	0.5	0.25	+	+
26	47.9	1.8	0.5	0.13	+	+
27	26.8	1.8	0.4	0	-	NP
28	37.4	1.5	0.5	0.11	+	+
29	77.0	1.5	0.7	0.14	+	+
30	25.1	1.7	0.4	0	-	NP
31	18.5	1.5	0.3	0.25	+	+
32	25.5	1.7	0.4	0.17	+	+
33	14.4	1.5	0.4	0	+	NP
34	18.7	1.6	0.3	0	+	NP
35	12.0	1.5	0.4	0.14	+	+
36	23.2	1.8	0.3	0.12	+	+
37	17.7	1.4	0.4	0.02	-	NP
38	12.8	1.7	0.5	0	-	NP

[]: concentration; A: absorbance; MSSs: mixed stained slides; +: performed with an adequate result; -: performed without an adequate result; NP: not performed.

**Table 6 diagnostics-13-01637-t006:** RNA extracted from MSSs of cytological samples.

RNA Cytological MSSs (*n*. 10 Slides)						
Case	[] (ng/µL)	A_260/280_	A_260/230_	Degree of Fragmentation	Molecular Test Success	
					Real Time PCR	NGS
1	11.5	1.4	0.4	0	-	NP
2	10.3	1.3	0.3	0	-	NP
3	43.2	1.4	0.4	0	-	NP
4	17.3	1.8	0.3	0	-	NP
5	10.7	1.2	0.3	0	-	NP
6	30.7	1.2	0.3	0	-	NP
7	71.6	1.2	0.5	0	-	NP
8	21.0	1.4	0.3	0	-	NP
9	18.7	1.7	0.2	0	-	NP
10	14.2	1.8	0.3	0	-	NP
11	25.8	1.6	0.3	0	-	NP
12	15.4	1.2	0.4	0	-	NP
13	25.7	1.6	0.3	0	-	NP
14	21.0	1.8	0.2	0	-	NP
15	22.0	1.2	0.3	0	-	NP
16	37.6	1.6	0.3	0	-	NP
17	11.1	1.6	0.5	0	-	NP
18	5.9	1.6	0.5	0	-	NP
19	9.1	1.5	0.3	0	-	NP
20	16.4	1.4	0.3	0	-	NP
21	9.8	1.5	0.2	0	-	NP
22	10.5	1.4	0.4	0	-	NP
23	9.6	1.3	0.4	0	-	NP
24	6.8	1.2	0.3	0	-	NP
25	6.2	1.2	0.5	0	-	NP
26	19.7	1.8	0.3	0	-	NP
27	54.8	1.4	0.4	0	-	NP
28	15.7	1.3	0.5	0	-	NP
29	59.8	1.5	0.6	0	-	NP
30	57.7	1.6	0.5	0	-	NP
31	40.0	1.6	0.4	0	-	NP
32	23.9	1.7	0.5	0	-	NP
33	16.0	1.8	0.5	0	-	NP
34	41.5	1.7	0.5	0	-	NP
35	26.4	1.8	0.3	0	-	NP
36	13.7	1.7	0.3	0	-	NP
37	41.8	1.7	0.5	0	-	NP
38	18.5	1.8	0.4	0	-	NP

[]: concentration; A: absorbance; MSSs: mixed stained slides; -: performed without an adequate result; NP: not performed.

## Data Availability

Not applicable.

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
