# Peer review of "A Critical Issue in Lung Cancer Cytology and Small Biopsies: DNA and RNA Extraction from Archival Stained Slides for Biomarker Detection through Real Time PCR and NGS—The Experience in Pathological Anatomy Unit"

_diagnostics, 2023, doi:10.3390/diagnostics13091637_

Round 1
Reviewer 1 Report
This manuscript describes the examination of DNA and RNA extraction from archival-stained slides. This is an important study because genetic information will become increasingly necessary. The overall manuscript is well described, but some minor points need to be reconsidered.
How many years ago did the author obtain these specimens? DNA and RNA degrade over time. Therefore, the samples were classified according to the date collected.
Were all specimens stored in the same condition? For example, was the HE stain discolored?
Author Response
Reviewer 1
This manuscript describes the examination of DNA and RNA extraction from archival-stained slides. This is an important study because genetic information will become increasingly necessary. The overall manuscript is well described, but some minor points need to be reconsidered.
How many years ago did the author obtain these specimens? DNA and RNA degrade over time. Therefore, the samples were classified according to the date collected. Were all specimens stored in the same condition? For example, was the HE stain discolored?
As correctly observed by the Reviewer, the storage condition, the time of archival, and the quality of HE staining could affect the DNA and RNA degradation. Thank you for the observation, since the year of collection and the storage condition represent a critical issue in the yield of nucleic acids extraction. Our samples were collected from 2019 to 2023, although the years of the collection were different, the condition of storage was the same for all samples (all stored in our archival) and the HE staining was not discolored. Our data showed no differences between the yield of nucleic acid extracted based on the years of collection. We added a supplementary table (Table S1) including the year of collection for each sample. Moreover, we improved also the Methods “Design of the study” as follows: “We retrospectively reviewed archival surgical samples, small biopsies, and cytological samples of lung adenocarcinoma with available material diagnosed at the Pathology Unit of the University of Campania Luigi Vanvitelli from 2019 to 2022. All samples were stored in the same condition in our archival.”
We hope our revisions will be adequate.

Reviewer 2 Report
In this manuscript, Zannini et al. reported a quantitative study of extracting DNA and RNA from archival tissue slides for NGS based molecular profiling of target genes. The authors performed DNA and RNA extraction experiments using the unstained slides and mixed stained slides from H&E, IHC, and FISH methods. They found that DNAs can be successfully extracted from most of the samples in the USs and the MSSs groups. Although RNA could also be extracted from both groups with sufficient amounts, the quality from the MSSs groups are highly fragmented and not suitable for further RT-PCR and NGS analyses. The design of this study is straightforward and is organized in good logic. 35 surgical samples were studied in this work. The DNA and RNA extraction and characterization experiments are carefully documented. However, the later NGS data analyses are oversimplified, which could be used in molecular profiling and biomarker detections, and supposed to be the major application of this work as suggested by the title and abstract. Without showing in depth NGS analyses and cancer related data from these pathological slides samples, the significance of this work will be greatly compromised.
Major concerns include:
1. Successful NGS data from these slides extractions should be provided, e.g. quality report of NGS data, total number of reads, gene assembly rates, sequencing depth of each gene, etc. This information will give readers an overview of if the slide extracted samples could provide reliable quality in NGSs for useful analyses and applications.
2. Gene Ontology analysis and gene pathway analyses should be performed, especially cancer related pathways, to support the claim that this strategy will be influential in cancer biomarker detections. A few key genes with significant expression level changes from the NGS analyses should be further validated by quantitative PCR or quantitative reverse transcription PCR to support the observations from the NGSs.
3. The RNA molecules are easily degraded if the storage condition is not optimal or with nuclease contaminations. It is interesting that none of the RNA extracts from MSSs passed the RT-PCR tests. For how long time have the MSSs been stored and what is the storage condition of the MSSs? Are the storage conditions and time the same between USs and MSSs groups?
Minor concerns:
4. How is the degree of fragmentation calculated? This information will be important to readers to evaluate the condition of their samples.
5. RT-PCR is the abbreviation of Reverse Transcription PCR rather than real time PCR. Thus, for DNA tests, the detection method in Tables 1, 3, 5 should be "real time PCR" or "qPCR" (quantitative PCR), or simply "PCR".
Author Response
Reviewer 2
In this manuscript, Zannini et al. reported a quantitative study of extracting DNA and RNA from archival tissue slides for NGS based molecular profiling of target genes. The authors performed DNA and RNA extraction experiments using the unstained slides and mixed stained slides from H&E, IHC, and FISH methods. They found that DNAs can be successfully extracted from most of the samples in the USs and the MSSs groups. Although RNA could also be extracted from both groups with sufficient amounts, the quality from the MSSs groups are highly fragmented and not suitable for further RT-PCR and NGS analyses. The design of this study is straightforward and is organized in good logic. 35 surgical samples were studied in this work. The DNA and RNA extraction and characterization experiments are carefully documented. However, the later NGS data analyses are oversimplified, which could be used in molecular profiling and biomarker detections, and supposed to be the major application of this work as suggested by the title and abstract. Without showing in depth NGS analyses and cancer related data from these pathological slides samples, the significance of this work will be greatly compromised.
Major concerns include:
1.Successful NGS data from these slides extractions should be provided, e.g. quality report of NGS data, total number of reads, gene assembly rates, sequencing depth of each gene, etc. This information will give readers an overview of if the slide extracted samples could provide reliable quality in NGSs for useful analyses and applications.
As correctly suggested by the Reviewer, the manuscript did not provide the NGS data thus our study lacks a critical point very important for the reader, particularly for a possible application in the practice of our approach. In this view, we improved our study with a supplementary table (Table S2) including mean coverage, uniformity of coverage, raw reads, and valid reads. We hope our revisions will be adequate.
- Gene Ontology analysis and gene pathway analyses should be performed, especially cancer related pathways, to support the claim that this strategy will be influential in cancer biomarker detections. A few key genes with significant expression level changes from the NGS analyses should be further validated by quantitative PCR or quantitative reverse transcription PCR to support the observations from the NGSs.
As correctly suggested by the Reviewer, the manuscript did not provide the NGS and Real time PCR data regarding the biomarkers that play a pivotal role in lung cancer. In this view, we improved our study including the results about EGFR and BRAF mutations that are currently approved as predictive biomarkers in lung cancer. We summarized these data in Table S2, including EGFR and BRAF status analyzed through NGS and Real Time PCR (the concordance between two assays), furthermore, we added the parameters of Real Time PCR.
Moreover, we improved the Methods “Real time PCR” in the main text as follows: “ΔCq is calculated as the difference between the Cq value of the gene mutation and the Cq value of the gene control. The sample is classified as mutated if the ΔCq value is equal or lower than the cut-off of the specific target, as reported in the manufacturer’s instructions.”
- The RNA molecules are easily degraded if the storage condition is not optimal or with nuclease contaminations. It is interesting that none of the RNA extracts from MSSs passed the RT-PCR tests. For how long time have the MSSs been stored and what is the storage condition of the MSSs? Are the storage conditions and time the same between USs and MSSs groups?
The RNAs extracted were stored at -80°C and the Real Time PCR analysis was performed within a week of the extraction to avoid the risk of degradation. All samples both USs and MSSs groups were managed under the same conditions. In this view, the quality of RNA extracted from the MSSs group is clearly influenced by factors due to previous ancillary analysis regardless of the storage.
Minor concerns:
- How is the degree of fragmentation calculated? This information will be important to readers to evaluate the condition of their samples.
As correctly observed by the Reviewer, we improved the Methods section explaining the calculation of the degree of fragmentation as follows: “The fragmentation was calculated as the ratio between the concentration (ng/µL) of the longer length and that of the shorter length amplicon of the nucleic acids. The degree of DNA fragmentation was calculated by Real Time PCR as the ratio between the amplification of longer fragments evaluated in the HEX channel and shorter fragments in the FAM channel. The degree of RNA fragmentation was calculated by Real Time PCR as the ratio between the amplification of longer fragments evaluated in the FAM channel and shorter fragments in the HEX channel.”
- RT-PCR is the abbreviation of Reverse Transcription PCR rather than real time PCR. Thus, for DNA tests, the detection method in Tables 1, 3, 5 should be "real time PCR" or "qPCR" (quantitative PCR), or simply "PCR".
Thank you for the observation. We modified the main text, Figure 1, and all Tables as marked.

Round 2
Reviewer 2 Report
The revisions are adequate. The quality of current manuscript meets the standard for acceptance.